# Feasibility of the Savvy Caregiver Program for LGBTQ+ Caregivers of People Living with Alzheimer’s Disease and Related Dementias

**DOI:** 10.3390/ijerph192215102

**Published:** 2022-11-16

**Authors:** Krystal R. Kittle, Rebecca Lee, Kiera Pollock, Yeonsu Song, Whitney Wharton, Joel G. Anderson, N. Maritza Dowling, Jason D. Flatt

**Affiliations:** 1Department of Social and Behavioral Health, School of Public Health, University of Nevada, Las Vegas, NV 89154, USA; 2School of Public Health, Boston University, Boston, MA 02118, USA; 3Los Angeles LGBT Center, Los Angeles, CA 90028, USA; 4School of Nursing, University of California, Los Angeles, CA 90095, USA; 5School of Nursing, Emory University, Atlanta, GA 30322, USA; 6College of Nursing, University of Tennessee Knoxville, Knoxville, TN 37996, USA; 7School of Nursing, George Washington University, Washington, DC 20006, USA

**Keywords:** caregiving, Alzheimer’s disease and related dementias, LGBTQ+, feasibility study

## Abstract

Nearly 350,000 lesbian, gay, bisexual, transgender, and queer/questioning (LGBTQ+) adults in the U.S. are currently living with Alzheimer’s disease and related dementias (ADRD). Informal caregivers face challenges impacting their ability to access and receive adequate and inclusive care for LGBTQ+ persons living with ADRD. The purpose of this study was to determine the feasibility and acceptability of the Savvy Caregiver Program for caregivers of LGBTQ+ individuals living with ADRD. Data for this secondary analysis come from caregivers (*n* = 17) who completed 6 sessions of the Savvy program. Caregivers were very satisfied with tailored program activities. Analyses of trends suggest non-significant increases in positive aspects of caregiving and decreases in caregiver burden and depressive symptoms. This is the first known study assessing the feasibility of the Savvy Caregiver Program for caregivers of LGBTQ+ individuals living with ADRD. Future research on the Savvy Caregiver Program for caregivers of LGBTQ+ people living with ADRD is needed.

## 1. Introduction

Alzheimer’s disease and related dementias (ADRD) are some of the most common health concerns faced by adults over the age of 65. In the United States, about 6.5 million older adults are living with ADRD, with projections of 14 million by 2060 [1]. ADRD is a progressive disease with common symptoms including memory loss, thinking and language problems that impact activities of daily living [1]. Due to the cognitive and physical effects of ADRD, people living with ADRD often rely on informal caregivers, such as family, friends and neighbors, who often lack any formal training in providing necessary care, such as nursing-related tasks. In 2021, informal caregivers provided an estimated 16 billion hours of unpaid care [1]. Informal care frequently allows individuals to age in place with dignity and to have better health outcomes [2]. However, caregivers often face a greater burden of depression, anxiety, and lower quality of life while providing this care [3,4].

Research suggests that lesbian, gay, bisexual, transgender, and queer/questioning (LGBTQ+) caregivers experience greater loneliness, financial strain, and poorer health than non-LGBTQ+ caregivers [5,6]. As a marginalized population, caregivers of LGBTQ+ older adults with ADRD face distinct challenges that may impact their ability to provide and access adequate and inclusive care for their care recipient, as well as affecting their willingness to access support services available for informal caregivers. For instance, many LGBTQ+ caregivers experience social support challenges, such as being more likely to live alone, not have children, and being single. Often, caregivers of LGBTQ+ older adults cannot rely on the care recipient’s biological family to assist with care, as many were rejected by family who were not accepting of their sexual orientation and/or gender identity [5,6]. LGBTQ+ older adults experience higher risks of disability, and LGBTQ+ caregivers and persons living with ADRD commonly experience discrimination when seeking health care and other aging-related social services [5,7,8]. Caregivers of LGBTQ+ older adults also experience significantly higher levels of lifetime depression, disability, victimization, discrimination, and stress [5,9,10,11,12].

There is currently limited research on the informal caregiving of LGBTQ+ older adults living with ADRD [11], and a dearth of studies regarding the acceptability of existing interventions for caregivers of LGBTQ+ older adults living with ADRD. Such research would allow for identifying innovative ways to recruit this marginalized and understudied population into ADRD caregiver interventions and research effectively. This research is needed to inform national efforts and health care providers and to shape future health care services, policies, and research that will ensure care is welcoming for all people [13,14]. Understanding the feasibility, acceptability, and future efficacy of a tailored Savvy Caregiver Program for caregivers of LGBTQ+ older adults with ADRD is one step toward addressing the psychosocial and health challenges experienced by this community.

The Savvy Caregiver Program is a widely used educational and behavioral program for caregivers of people living with ADRD [15,16]. The Savvy training course provides skills and knowledge needed to care for someone with ADRD efficiently by improving caregiver mastery, increasing social support, and reducing the adverse impacts of caregiving. Adapted from the Minnesota Family Workshop, the Savvy Caregiver Program formalized the curriculum, created comprehensive training materials, and made additional multimedia additions [16]. To ensure cultural acceptability, Savvy has been adapted for diverse populations, including rural, veteran, and racial/ethnic minority (i.e., Hispanic/Latino, Black/African American, and Asian/Pacific Islander communities) caregivers [5,9,10,11,12]. However, there is limited research on the cultural acceptability and feasibility of the Savvy Caregiver Program for caregivers of LGBTQ+ people living with ADRD. The purpose of the current study was to determine the feasibility and acceptability of the adapted Savvy Caregiver Program and explore trends in caregiver outcomes among those who completed the program.

## 2. Materials and Methods

### 2.1. Design

This secondary data analysis explored the feasibility and acceptability of a culturally adapted Savvy Caregiver Program designed to support caregivers of LGBTQ+ older adults living with ADRD [17]. First, three complete Savvy Caregiver Program sessions, comprised of six, two-hour classes, were delivered to 18 caregivers of LGBTQ+ older adults living with ADRD. We were able to collect data from 17 participants, but only 11 participants (65%) provided complete data on both the pre- and post-intervention surveys. Data were used to analyze the feasibility and acceptability of the adapted program and examine trends in common caregiver-related outcomes used in the Savvy Caregiver Program. Given this study involved secondary data analysis, it was considered exempt by the University of California, San Francisco Human Research Protection Program Institutional Review Board.

### 2.2. Savvy Caregiver Program Adaptation and Content

The Savvy Caregiver Program is comprised of six two-hour sessions. Each session focuses on different aspects of caregiving for individuals with ADRD. Session themes include, for example, general knowledge about ADRD, creating caregiving plans, and creating personal resources to support caregiving [15]. Sessions involve a mix of didactic tools, including teaching sessions, role playing, videos, and group discussion. The sessions were held once a week in-person at a local community-based organization providing services for LGBTQ+ older adults. Three separate groups of caregivers completed all the six sessions of the adapted Savvy Caregiver Program.

Prior to adapting and implementing the Savvy Caregiver Program for the LGBTQ+ community, a community-based organization’s staff had interviews with 10 current caregivers of LGBTQ+ people living with ADRD. The interview questions focused on caregivers’ backgrounds, relationships with their care recipients, challenges in caregiving, issues specific to the LGBTQ+ community (e.g., non-traditional caregiving roles such as friends and neighbors), ability to meet caregiving needs, supports currently received, additional services needed, willingness to participate in caregiver support groups, description of the program and assessing program interest, and identification of barriers and facilitators to participation in the six-week Savvy Caregiver Program. One interesting finding that came out of the consultations was around language and a recommendation to consider using the term care partner rather than caregiver. It was suggested that the term, “care partner”, reflects a more egalitarian relationship between the person providing care and the care recipient, which could be important for addressing broader concerns related to stigma and assumptions about caregiving burden. Interviews also identified ways to recruit LGBTQ+ caregivers, including advertising to ADRD support groups, LGBTQ+ aging services and programs, and referrals from therapist and clinicians. Findings from the interviews with caregivers were also used to adapt some of the existing scenarios and role-playing activities that are part of the Savvy Caregiver Program curriculum. For example, a scenario was developed to represent how LGBTQ+ persons living with ADRD often live alone and LGBTQ+ caregivers may be non-relatives, friends or a neighbor. In this scenario, participants problem solved around keeping the person living alone with ADRD safe, engaged in activities (e.g., movies, art, and other hobbies), as well as supporting meal planning. Finally, given described limitations in access to respite care, LGBTQ+ caregivers recommended that the program provide space and respite services, such as a planned activity for persons living with ADRD at the same time as the Savvy Caregiver Program so that LGBTQ+ caregivers could attend the weekly sessions and not have to find alternative care support.

### 2.3. Feasibility and Acceptability of the Adapted Program

To assess feasibility and acceptability of the adapted program, data was collected at the last session, after participants completed the 6-week program. Participants were asked to respond to seven questions (1 = strongly disagree to 5 = strongly agree) about satisfaction with and acceptability of the adapted Savvy Caregiver Program. These questions were used in a previous study and assessed satisfaction with program content, knowledge and skills gained, appropriateness of the training in increasing knowledge of ADRD and caregiving, personal confidence, feedback on the number of sessions for the program, and whether they would recommend the program to others [16].

### 2.4. Measures

Demographic data included age, race/ethnicity, gender identity, sex assigned at birth, sexual orientation, level of educational attainment, income, and relationship to the care recipient. We utilized existing measures currently used to test the efficacy of the Savvy Caregiver Program [15,16,17,18,19,20,21]. To minimize time and burden with data collection, we used abbreviated scales whenever possible. Pre- and post-intervention surveys included the 12-item Short-Form Zarit Caregiver Burden Index, 4-item Center for Epidemiologic Studies Depression Scale (CES-D), 7-item caregiver mastery scale, and the 11-item positive aspects of caregiving [20,21,22,23,24]. The Zarit Caregiver Burden Index was scored based on a 5-point Likert-type scale (1 Never to 5 Nearly always), with total scores ranging from 12 to 60. The CES-D measures depressive symptoms during the past week ranging from (1) rarely or none of the time (less than 1 day), (2) some or a little of the time (1–2 days), (3) occasionally or a moderate amount of the time (3–4 days), or (4) most or all of the time (5–7 days), with total scores ranging from 4 to 16. For the Caregiver Mastery Scale, questions were scored on a Likert scale from (4) strongly agree to (1) strongly disagree, with three negative statements being reverse-scored. Total scores range from 7 to 28, with higher scores reflecting greater caregiver mastery [18,22]. The Positive Aspects of Caregiving Scale was scored from (5) agree a lot to (1) disagree a lot, and total scores ranged from 11 to 55; higher scores reflected more positive aspects of caregiving. An additional seven satisfaction questions were included in the post-survey to assess acceptability of the Savvy program [16]. The pre- and post-intervention surveys took about 30 min on average to complete.

### 2.5. Sample and Setting

Caregivers were recruited through advertising by the local LGBTQIA+ community center and through case managers. A community-based event coordinator publicized the availability of the program through social media, newsletters, emails, and leaflets. Case managers recruited through caregiver support groups, current caregiver clients, and by reaching out to any LGBTQ+ clients’ living with ADRD who currently had a caregiver. To enroll in the program, participants emailed or called the program administrator to confirm their availability and ability to attend and complete the six-week program.

### 2.6. Data Analysis

Secondary analysis of data regarding feasibility and related quantitative data from pre- and post-intervention surveys were administered in-person by the program staff (K.P.) and entered into an online survey and data management program. Results are presented for both the matched (i.e., participants who completed both the pre- and post-intervention surveys) and the unmatched sample (i.e., participants who completed either the pre- or post-intervention survey). The data were analyzed in R using non-parametric descriptive statistics, such as frequencies, mean and median scores. To look at potential feasibility and impact of the program on caregiver outcomes, we used Mann–Whitney tests and Wilcoxon signed-rank tests to assess differences before and after completion of the program for independent groups and matched pairs (*n* = 11). All tests were conducted using the alpha level of 0.05.

## 3. Results

### 3.1. Demographic Characteristics

A total of seventeen participants provided demographic information (Table 1) in either the pre- or post-intervention surveys.

The median age of participants was 66.0 years (range: 52–82 years; IQR: 14.0). Among participants, two-thirds identified as gay and one-third identified as straight. One participant identified as non-binary. More than half of the participants identified as White (56.3%), with Hispanic/Latino (25%), Asian/Asian-American (12.5%), and Black/African American (6.3%) making up the remainder. Two participants (13.3%) reported a positive HIV status with the rest of participants reporting negative or unknown status. Most participants (94.1%) were not currently working, with only one participant reporting they worked part-time. About 70% of participants reported they had more than a high school level education and 80% reported an income of $10,000 or less per year.

For caregiving-related background, we examined the care partner’s relationship to the care recipient, the main reasons for providing care, and the length of time providing care. Regarding relationship type, half of respondents provided a write-in response. Nearly half of participants reported caring for non-relatives or friends, and over half reported caring for a spouse or partner (25%) or other relative (31%). Additional write-in responses included “not a care provider” and “neighbor.” When asked about the main reasons for providing care, 64.71% of participants reported ADRD, 35.29% general frailty, and 29.41% physical illness. Approximately half of participants provided care for more than one reason. Most participants (62.64%) had provided care for fewer than five years, while the remaining participants had provided care for ten or more years.

### 3.2. Unmatched Survey Results

Notable trends in outcomes after completing the intervention were found when comparing the pre- and post-intervention survey responses among the unmatched sample (Table 2; *n* = 11). There was a small increase in the median response to the caregiver burden questions between the pre- (median = 2.58; IQR = 1.54) and post-intervention (median = 2.92; IQR = 0.67) survey responses. For depressive symptoms (i.e., CES-D), there was a small decrease between the pre-intervention median of 1.63 (IQR = 1.13) and the post-intervention median of 1.50 (IQR = 0.75). However, there were increases in positive aspects of caregiving from pre-intervention (median = 3.91; IQR = 1.18) to post-intervention (median = 4.27; IQR = 1.14). For caregiver mastery, there was no change in overall scores from pre-intervention (median = 16.00; IQR = 3.50) to post-intervention score (median = 16.00; IQR = 3.75).

### 3.3. Matched Survey

The next analysis used a paired dataset of participants who completed both the pre- and post-intervention surveys (Table 3). Sample sizes with complete data varied between the Zarit Caregiver Burden Index (*n* = 6), CES-D (*n* = 8), Caregiving Mastery (*n* = 9) and Positive Aspects of Caregiving (*n* = 7) scales. For both the Zarit Caregiver Burden Index and the CES-D, there was a slight decrease in median score. The median score on the Zarit Caregiver Burden Index at pre-intervention was 3.08 (IQR = 1.75) and 2.75 (IQR = 0.69) at post-intervention. The CES-D pre-intervention score was 2.00 (IQR = 1.63) and 1.50 (IQR = 0.38) at post-intervention. There was an increase in median score of the positive aspects of caregiving from 3.91 (IQR = 0.91) pre-intervention to 4.25 (IQR = 1.09) post-intervention. In terms of caregiver mastery, there was a slight decrease from pre-intervention (median = 17.00, IQR = 3.00) to post-intervention (median = 16.00, IQR = 4.00). Given the small sample size, we did not have power to exam statistical significance.

### 3.4. Feasibility and Acceptability Findings

At the conclusion of the Savvy Caregiver Program, participants were asked about their overall satisfaction with the program (Table 4). Of the 17 unique survey respondents, 12 completed the satisfaction survey (71%). Most of these participants reported feeling more knowledgeable about ADRD care (Median 5.00), and that they would recommend the training to other caregivers (Median 5.00). 

## 4. Discussion

Findings from this feasibility study of the Savvy Caregiver Program for caregivers of LGBTQ+ persons living with ADRD suggest that LGBTQ+ populations would benefit from caregiver interventions and related strategies to support those caring for someone living with ADRD. The data collected from this study mirror findings from studies with larger sample sizes and diverse caregivers [18,20,25]. For instance, the high satisfaction rate with the program may suggest the program is feasible and acceptable for caregivers of LGBTQ+ persons living with ADRD. These feasibility findings could help to inform future iterations and tailoring of the Savvy Caregiver Program for the caregivers of LGBTQ+ persons living with ADRD. This includes considering respite care services and tailoring of scenarios for the LGBTQ+ community, such as including non-relative caregivers and caregiving scenarios for care recipients that live alone. Additionally, it may be important to tailor to the skill level of caregivers, the unique caregiver/care recipient relationship dynamics (non-relatives, friends, and other part-time caregivers), and to provide LGBTQ+ inclusive care resources.

While our study was underpowered to find significant differences in measures related to the efficacy of the intervention, we found modest improvements in several of the key measures used to assess the impact of the Savvy Caregiver Program. Pre- and post-intervention data suggest trends in reduction of caregiver burden and depressive symptom and improvements related to positive aspects of caregiving. A study testing the Savvy Caregiver Program for racial/ethnic minority caregivers (Hispanic/Latino, Black/African American, and Asian/Pacific Islanders) found similar results in terms of reducing depressive symptoms [16]. This study also found improvements in overall caregiver mastery; however, we found that caregiver mastery remained relatively stable, which may be due to our smaller sample size. There is a need for further examination of caregiver mastery, skill development, and related caregiver measures among caregivers of LGBTQ+ persons living with ADRD. Further research is also needed to assess and intervene on caregivers’ skills in providing care to LGBTQ+ people living with ADRD.

Participants reported high satisfaction with all aspects of the program. High satisfaction with Savvy Caregiver Program has been widely established, with several studies finding that caregivers rate the program as helpful [25,26]. A study with rural caregivers who completed Savvy found these caregivers were highly satisfied regardless of the number of sessions and that the program helped them to identify social support services [26]. Another statewide training program of the Savvy Caregiver Program in Michigan, which involved a train-the-trainer model, found high satisfaction among both trainers and caregivers, with nearly 5000 participants reporting they would recommend the training to other ADRD caregivers [27].

There are several areas that should be considered in future studies in terms of the feasibility of the Savvy Caregiver Program for caregivers of LGBTQ+ people living with ADRD. First, research has shown LGBTQ+ caregivers are more likely to live in separate households their care recipients [5]. Additionally, caregiver relationships likely vary for LGBTQ+ caregivers, with one in 10 identifying as a friend or neighbor [6]. Future tailoring and adaptations of the Savvy Caregiver Program for LGBTQ+ caregivers should consider incorporating non-traditional caregiving roles, such as by those who may provide part-time or less frequent care based on when it is needed, into the current curriculum, scenarios, and role-playing activities. Additionally, very little is known about those that utilize care circles (i.e., a care support network) as well as the need to explore experiences of transgender caregivers of people living with ADRD. A previous study with LGBTQ+ ADRD caregivers found nearly 20% of caregivers reported caring for a transgender person living with ADRD and 13% of the caregivers identified as transgender [6]. Thus, there is a need to understand how the Savvy Caregiver Program can meet the needs of both transgender caregivers and transgender people living with ADRD. Additionally, research on how the Savvy Caregiver Program impacts health outcomes and experiences of LGBTQ+ people living with ADRD should be considered. Finally, researchers might consider the language used when referring to those providing care by using more egalitarian terminology, such as “care partner”, to help reduce the stigma associated with the potential burden of providing care to someone living with ADRD.

There are several limitations of this feasibility study that should be considered. First, our small sample size and challenges related to both pre- and post-intervention survey completion limit the generalizability of our findings. Additionally, sample size was not predetermined to ensure power as we conducted a secondary data analysis of a community implemented program. However, the purpose of the study was to examine feasibility of the Savvy Caregiver Program and was not intended to determine efficacy or effectiveness. This study was a secondary data analysis, so some of the details on feasibility and reasons for non-response to pre- and post-intervention surveys could not be determined. For instance, more information on acceptability, such as the number of participants contacted to participate who did not end up enrolling, should be considered. A total of four participants or 24% did not complete the post-test at program completion and were lost to follow up. Thus, future research should consider the overall acceptance and dropout rates for the Savvy Caregiver Program. There is also the possibility for non-response bias among the individuals who only completed either the pre- or post-intervention surveys, and the potential that social desirability bias may have impacted scores on satisfaction with the program. Future randomized controlled studies testing Savvy with caregivers of LGBTQ+ people living with ADRD should be considered. Additionally, there were some unique demographic characteristics of our participants, with very few participants reporting current employment and most participants being over the age of 65. Employed caregivers might experience greater barriers in terms of attendance of the required sessions, as well as greater challenges in balancing caregiving demands. Research suggests LGBTQ+ caregivers tend to be younger than their heterosexual and cisgender counterparts [7]. Thus, future studies should consider recruiting diverse caregivers and examining the feasibility of the Savvy Caregiver Program with younger LGBTQ+ caregivers. However, the demographic characteristics of study participants in our study were reflective of participants who utilized services and participated in LGBTQ+ aging programming in the geographic region.

## 5. Conclusions

This work represents the first to examine the feasibility of the Savvy Caregiver Program for LGBTQ+ caregivers of people living with ADRD. There is a need for additional research to promote culturally relevant aging services and behavioral interventions for LGBTQ+ caregivers and their care recipients living with ADRD. For example, future studies examining the feasibility and efficacy of the Savvy Caregiver Program for LGBTQ+ caregivers should consider their diverse backgrounds including age, sexual orientation, gender identity, race/ethnicity, and employment status. This also includes exploring relationship types (e.g., spouse/partner, friends or neighbors) between LGBTQ+ caregivers and people living with ADRD, and understanding the needs of LGBTQ+ caregivers in terms of knowledge, skills, and health outcomes. Finally, there is a need for additional programming and services addressing the unique health and psychosocial needs of LGBTQ+ caregivers and LGBTQ+ people living with ADRD.

## Figures and Tables

**Table 1 ijerph-19-15102-t001:** Demographic Characteristics of Caregivers.

Characteristic	% (*n*)
Age, Median, (IQR)	66.00 (14.00)
Race (*n* = 16)	
Black/African American	6.25 (1)
Asian/Asian-American	12.50 (2)
Hispanic/Latino	25.00 (4)
White	56.25 (9)
Gender (*n* = 17)	
Female	29.41 (5)
Male	64.71 (11)
Non-binary	5.88 (1)
Sex Assigned at Birth (*n* = 17)	
Male	70.59 (12)
Female	29.41 (2)
Sexual Orientation (*n* = 15)	
Gay/Lesbian	66.67 (10)
Heterosexual/Straight	33.33 (5)
HIV Status (*n* = 15)	
Positive	13.33 (2)
Negative	73.33 (11)
Don’t Know	13.33 (2)
Disability Status (*n* = 16)	
Yes	6.25 (1)
No	93.75 (15)
Employment (*n* = 17)	
Employed fewer than 40 h per week	5.88 (1)
Not currently employed	94.12 (16)
Education (*n* = 13)	
High school or less	30.77 (4)
Some college	23.08 (3)
4-year college	15.38 (2)
Graduate School	30.77 (4)
Income (*n* = 10)	
Less than $10 K	20 (2)
$10–$25 K	40 (4)
$50 K+	40 (4)
Relationship to Care Recipient (*n* = 16)	
Non-relative or friend	43.75 (7)
Spouse or Partner	25.00 (4)
Other relative	31.25 (5)
Reasons for providing care (*n* = 17)	
Physical Illness	29.41 (5)
General Frailty	35.29 (6)
Mental Health Issues	23.53 (4)
Dementia/Confusion	64.71 (11)
Reassurance	23.53 (4)
Other	11.76 (2)
More than one reason for providing care	52.94 (17)
Length of time providing care (*n* = 11)	
0–5 years	63.64 (7)
10+ years	36.36 (4)

Note: IQR is interquartile range.

**Table 2 ijerph-19-15102-t002:** Measures Assessed Before and After Savvy Caregiver Program (Unmatched sample).

Measure	Pre-Test Median (IQR)	Post-Test Median (IQR)
Zarit Caregiver Burden Index, *n* = 11	2.58 (1.54)	2.92 (0.67)
CES-D, *n* = 13	1.63 (1.13)	1.50 (0.75)
Positive Aspects of Caregiving, *n* = 11	3.91 (1.18)	4.27 (1.14)
Caregiver Mastery, *n* = 13	16.00 (3.50)	16.00 (3.75)

Note: IQR is interquartile range; CES-D, Center for Epidemiologic Studies Depression Scale.

**Table 3 ijerph-19-15102-t003:** Measures Assessed Before and After Savvy Caregiver Program (Matched).

Measure	Pre-Test Median (IQR)	Post-Test Median (IQR)
Zarit Caregiver Burden Index, *n* = 6	3.08 (1.75)	2.75 (0.69)
CES-D, *n* = 8	2.00 (1.63)	1.50 (0.38)
Positive Aspects of Caregiving, *n* = 7	3.91 (0.91)	4.25 (1.09)
Caregiver Mastery, *n* = 9	17.00 (3.00)	16.00 (4.00)

Note: IQR is interquartile range; CES-D, Center for Epidemiologic Studies Depression Scale.

**Table 4 ijerph-19-15102-t004:** Satisfaction and Feasibility of the Savvy Program (*n* = 12).

Question	Median (IQR)
I feel the content of the Savvy Caregiver Program was relevant to my situation.	5.00 (0.00)
I feel I learned useful strategies for providing care to people with dementia.	5.00 (1.00)
I feel more knowledgeable about dementia care.	5.00 (0.00)
I feel I have more skills.	5.00 (0.00)
I feel more confident in myself.	5.00 (1.00)
I would recommend the Savvy Caregiver program to others providing care to those with dementia.	5.00 (0.00)
I feel the number of sessions was appropriate.	5.00 (0.00)

Note: IQR is interquartile range.

## Data Availability

The data presented in this study are available on request from the corresponding author. The data are not publicly available to protect confidentiality of the research participants.

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
