# Peer review of "Feasibility of the Savvy Caregiver Program for LGBTQ+ Caregivers of People Living with Alzheimer’s Disease and Related Dementias"

_ijerph, 2022, doi:10.3390/ijerph192215102_

Round 1

Reviewer 1 Report

It is great pleasure to see that the authors have put effort to investigate the topic. I think that it is really important to understand the program effectiveness and feasibility among the sexual minority group. Therefore, I admire the present study, although the sample size was very small. However, there are some concerns and I invite the authors to clarify.

1. I think that the first major problem in the present contribution is that it is unclear if the authors targeted on LGBTQ+ older people with AD, or the authors targeted on LGBTQ+ caregivers of older people with AD. The authors said, "inclusive care for LGBTQ+ persons living with ADRD" and "caregivers of LGBTQ+ individuals living with ADRD" in the Abstract. The former gives the impression that the authors focused on people with AD who are LGBTQ+, while the latter indicates that the authors focused on caregivers who are LGBTQ+. This kind of confusing statements can be found throughout the manuscript. Or, do the authors meant that they want to investigate the topic when both caregivers and older people with AD are LGBTQ+? I think that the authors should make this clear at the very beginning of the Introduction and Abstract.

2. Following the previous comment, I only see the demographic information of the caregivers but not the older people with AD. I think that it is important to have the information regarding the care recipients. 

3. I wonder if the Savvy Caregiver Program for LGBTQ+ caregivers has been modified from the original Savvy Caregiver Program. Or, are they totally the same programs?

4. It is important to report how many caregivers were approached to receive the Savvy Caregiver Program for LGBTQ+ caregivers. Specifically, this is a feasibility study and it is important to know whether the acceptance rate in receiving such a program is high. If the acceptance rate is low, the authors have to discuss why not many caregivers do not want to join this program.

5. Another important information for the feasibility study is the drop-out rate. That is, how many caregivers did not complete the entire Savvy Caregiver Program for LGBTQ+ caregivers? This information is important for healthcare providers to know if this program is too demanding.

6. The authors said that they used secondary data analysis. Then, I wonder if there are other publications using the same dataset? 

Author Response

Response: We thank the reviewer for their thoughtful comments, which have helped us to strengthen the paper. Please see our point-by-point responses below. All page numbers referenced are for the tracked changes version of the revised manuscript.

  1. I think that the first major problem in the present contribution is that it is unclear if the authors targeted on LGBTQ+ older people with AD, or the authors targeted on LGBTQ+ caregivers of older people with AD. The authors said, "inclusive care for LGBTQ+ persons living with ADRD" and "caregivers of LGBTQ+ individuals living with ADRD" in the Abstract. The former gives the impression that the authors focused on people with AD who are LGBTQ+, while the latter indicates that the authors focused on caregivers who are LGBTQ+. This kind of confusing statements can be found throughout the manuscript. Or, do the authors meant that they want to investigate the topic when both caregivers and older people with AD are LGBTQ+? I think that the authors should make this clear at the very beginning of the Introduction and Abstract.

Response: We appreciate this suggestion. We have clarified in the abstract and introduction that the focus was on caregivers. We revised in the manuscript to emphasize the focus on caregivers vs. LGBTQ+ people living with ADRD.

  1. Following the previous comment, I only see the demographic information of the caregivers but not the older people with AD. I think that it is important to have the information regarding the care recipients. 

Response: We appreciate this comment. Unfortunately, no data was collected on the person living with ADRD. We have noted this as a limitation on page 8 lines 313-315

“Additionally, research on how the Savvy Caregiver Program impacts health outcomes and experiences of LGBTQ+ people living with ADRD should be considered.”

  1. I wonder if the Savvy Caregiver Program for LGBTQ+ caregivers has been modified from the original Savvy Caregiver Program. Or, are they totally the same programs?

Response: We appreciate this comment. We highlight a bit on page 3 (lines 107-133) that interviews were conducted with 10 current caregivers of LGBTQ+ people living with ADRD. We have revised this section to clarify the interviews and key findings.

  1. It is important to report how many caregivers were approached to receive the Savvy Caregiver Program for LGBTQ+ caregivers. Specifically, this is a feasibility study and it is important to know whether the acceptance rate in receiving such a program is high. If the acceptance rate is low, the authors have to discuss why not many caregivers do not want to join this program.

Response: Unfortunately, we did not have data on the number of participants contacted that did not participate. We have added a statement to our limitations to emphasize the need to explore this as an important indicator of feasibility. Please see page 9 lines 325-329.

“For instance, more information on acceptability, such as the number of participants contacted to participate who did not end up participating and the number who did not complete the entire program were not captured. This data could provide more insight into the overall acceptance rate for the program.”

  1. Another important information for the feasibility study is the drop-out rate. That is, how many caregivers did not complete the entire Savvy Caregiver Program for LGBTQ+ caregivers? This information is important for healthcare providers to know if this program is too demanding.

Response: Please see previous comment where this concern was addressed in the limitations.

  1. The authors said that they used secondary data analysis. Then, I wonder if there are other publications using the same dataset? 

Response: We did previously present preliminary data at an academic conference, Alzheimer’s Association International Conference, but no other publications occurred with this dataset.

Reviewer 2 Report

The manuscript is well structed and here few suggestions which further improve the quality of the manuscript. 

1. Conclusion should be concise with the abstract.

2. Results can be improved by using graphical presentations

3. There should be more detail provided about Alzheimer’s Disease in Introduction session (small paragraph).  

Author Response

Response: We thank the reviewer for their thoughtful comments, which have helped us to strengthen the paper. Please see our point-by-point responses below. All page numbers referenced are for the tracked changes version of the revised manuscript.

  1. Conclusion should be concise with the abstract.

Response: Thank you for this suggestion. We have revised the abstract to make conclusion more concise.

  1. Results can be improved by using graphical presentations

Response: Thank you for this suggestion. We have revised results and tables based on reviewers’ feedback to ensure greater clarity.

  1. There should be more detail provided about Alzheimer’s Disease in Introduction session (small paragraph).  

Response: We appreciate this suggestion. We have added the following statement to the introduction to provide more details about Alzheimer’s disease

“ADRD is a progressive disease with common symptoms including memory loss, thinking and language problems that impact activities of daily living [1].”

Reviewer 3 Report

This is an excellent study that highlights what the Savvy Caregiver Program means and how acceptable it is to those caring for LGBTQ+ people with ADRD.

The manuscript is well written.

1. How was the sample size for this study determined? Please provide the rationale for this decision.

2. Calculate the effect size, not just the P-value.

Author Response

Response: We thank the reviewer for their thoughtful comments, which have helped us to strengthen the paper. Please see our point-by-point responses below. All page numbers referenced are for the tracked changes version of the revised manuscript.

  1. How was the sample size for this study determined? Please provide the rationale for this decision.

Response: Sample size unfortunately was not determined prior to the study. This study relied on secondary data from a community organization that adapted and implemented the program. We have added this to the limitation. We added the following statement to limitations: “Additionally, sample size was not predetermined to ensure power as we conducted a secondary data analysis of a community implemented program.”

  1. Calculate the effect size, not just the P-value.

Response: We appreciate the reviewer’s recommendation. However, effect sizes in studies with small samples are more highly variable. We are concerned that with our n=11 to 13 we may overestimate effect size, which could result in low reproducibility of results in future studies. We based our decision on the following article: Ioannidis, J. P. Why most published research findings are false. PLoS Med. 2, e124 (2005) and guidance from a statistician on our team.

Round 2

Reviewer 1 Report

The authors have responded to my previous comments to improve the work. However, some revisions are needed.

1. I asked about drop-out rate, but the authors did not respond this. They refer to the response on the response rate. Drop-out rate and response rate are different things. Drop-out means that the person enters a program or an interview but does not complete the whole process; response rate means that the person does not take part after invitation. I can understand that the authors could not have the data regarding response rate. However, I cannot understand that the authors do not have the data of drop-out rate.

2. I think that the authors should cite their prior conference paper in the reference list. 

Author Response

1. I asked about drop-out rate, but the authors did not respond this. They refer to the response on the response rate. Drop-out rate and response rate are different things. Drop-out means that the person enters a program or an interview but does not complete the whole process; response rate means that the person does not take part after invitation. I can understand that the authors could not have the data regarding response rate. However, I cannot understand that the authors do not have the data of drop-out rate.

Response: We apologize for not fully addressing the reviewer's comments about drop out rate. We still do not have complete information on drop out rate, however, we know that only 13 out of the 17 completed the post-test surveys, so potentially 4/17 or 24% did not complete the program. We have revised our limitations to reflect this concern.

“For instance, more information on acceptability, such as the number of participants contacted to participate who did not end up enrolling should be considered. A total of four participants or 24% did not complete the post-test at program completion and were lost to follow up. Thus, future research should consider the overall acceptance and dropout rates for the Savvy program.”

2. I think that the authors should cite their prior conference paper in the reference list. 

Response: We have added the citation.

  1. Flatt, J. D.; Pollock, K.; Lee, R.; Song, Y.; Wharton, W.; Anderson, J. G. Feasibility of the Savvy Caregiver program for care providers of LGBTQ adults living with Alzheimer’s disease and related dementias. Alzheimer's & Dementia 2021, 17, e055633. https://doi.org/10.1002/alz.055633